# Elimination of noise in optically rephased photon echoes

You-Zhi Ma[1,2], Ming Jin[1,2], Duo-Lun Chen[1,2], Zong-Quan Zhou [1,2 ✉], Chuan-Feng Li [1,2 ✉] & Guang-Can Guo[1,2]

Photon echo is a fundamental tool for the manipulation of electromagnetic fields. Unavoidable spontaneous emission noise is generated in this process due to the strong rephasing pulse, which limits the achievable signal-to-noise ratio and represents a fundamental obstacle towards their applications in the quantum regime. Here we propose a noiseless photon-echo protocol based on a four-level atomic system. We implement this protocol in a $Eu^{3+}:Y_2SiO_5$ crystal to serve as an optical quantum memory. A storage fidelity of $0.952 \pm 0.018$ is obtained for time-bin qubits encoded with single-photon-level coherent pulses, which is far beyond the maximal fidelity achievable using the classical measure-and-prepare strategy. In this work, the demonstrated noiseless photon-echo quantum memory features spin-wave storage, easy operation and high storage fidelity, which should be easily extended to other physical systems.

[1] CAS Key Laboratory of Quantum Information, University of Science and Technology of China, Hefei, China. [2] CAS Center for Excellence in Quantum Information and Quantum Physics, University of Science and Technology of China, Hefei, China. ✉email: zq_zhou@ustc.edu.cn; cfli@ustc.edu.cn

Echoes are coherent emissions from atoms when they are interacting with a series of electromagnetic pulses. This phenomenon was firstly discovered by Erwin Hahn in 1950[1] in the radio-frequency (RF) domain and historically it is named as spin echo. The underline physics between spin-echo and photon echo (PE)[2] are the same: a strong electromagnetic pulse rephase an inhomogeneously broadened atomic ensemble so that the initial excitation will be refocused in a specific time.

PE has shown great capabilities in storage and manipulation of input photons, which attracts much attention in the microwave (MW) regime since it enables efficient interfacing with super-conducting quantum processors[3,4], and in the optical regime since it serves as a building block for the memory-based large-scale quantum networks[5,6]. However, the population-inverted medium produced by the rephasing pulse generates strong spontaneous emission noise, which represents a fundamental limit to the achievable signal-to-noise ratio and prevents PE from directly working in the quantum regime[7]. Bringing PE to the quantum regime has been a long-standing challenge with wide-spread applications[8,9].

In the optical regime, controlled reversible inhomogeneous broadening[10–12] and atomic frequency comb (AFC)[13–16], successfully avoid such noise by abandoning the optical rephasing pulse, at the expense of reduced sample absorption after a complex spectral preparation step which typically leads to a reduced storage efficiency[17]. Such protocols are challenging to be extended to other frequency bands since spectral-hole burning is required to tailor the natural atomic absorption. To solve this problem, the revival of silenced echo (ROSE) has been proposed to double rephase the atomic ensemble to avoid population inversion and to make use of the natural absorption[18]. However, experimental implementations of such protocols in the quantum regime have been proven to be extremely challenging, since a slight imperfection of rephasing pulses will lead to the residual population in the excited state, which generates indistinguishable spontaneous emission noise[19]. On the other hand, four-level photon echo (4LE) has been proposed to effectively rephase the atomic ensemble with two $\pi$ pulses at different frequencies as compared to that of the input so that the coherent noise can be suppressed by frequency filtering[20]. However, the echo still emits in a population-inverted medium with strong spontaneous emission noise. As a result, whether PE with optical rephasing can operate in the single-photon regime remains elusive to date.

Here, inspired by 4LE and ROSE, we propose a noiseless photon-echo (NLPE) protocol that can eliminate both the coherent noise and the spontaneous emission noise, based on double rephasing in a four-dimensional atomic Hilbert space. We experimentally implement this protocol in a $Eu^{3+}:Y_2SiO_5$ crystal which is a unique material that enables coherent optical storage for hours[17,21]. A signal-to-noise ratio above 40 is obtained for single-photon level input, which facilitates high-fidelity quantum storage of time-bin qubits.

## Results

**Experimental setup**. We will introduce this protocol based on the actual level structure of our experimental sample (Fig. 1a) but we note that a four-level atomic system will be sufficient. The memory crystal is a $^{151}Eu^{3+}:Y_2SiO_5$ crystal with a concentration of 0.01% and a length of 8 mm along the crystallographic $b$-axis. To achieve noise suppression in the frequency domain, a 0.1% $^{151}Eu^{3+}:Y_2SiO_5$ crystal with a length of 15 mm along the $b$-axis is employed as the filter crystal.

Since the inhomogeneous broadening of the memory crystal (0.7 GHz) is much larger than the hyperfine splittings, a preparation procedure is employed to isolate a well-defined

four-level system in the memory crystal for implementing our NLPE protocol (Fig. 1b). This preparation procedure is not a part of NLPE protocol since it is not required if the inhomogeneous broadening is smaller than the hyperfine level splittings[22] or working in the RF/MW regime. Four energy levels are involved in this protocol, where 1 denotes $\left|\pm 1/2\right\rangle_g$, 3 denotes $\left|\pm 3/2\right\rangle_g$, $\bar{3}$ denotes $\left|\pm 3/2\right\rangle_e$ and $\bar{5}$ denotes $\left|\pm 5/2\right\rangle_e$, respectively. Without causing confusion, we use $f_{ij}$ to denote the frequency of light, which is resonant with $\left|\pm i/2\right\rangle_g \leftrightarrow \left|\pm j/2\right\rangle_e$ atomic transition. The first step is the so-called class cleaning process where four pump pulses with center frequencies of $f_{15}$, $f_{35}$, $f_{13}$ and $f_{53}$ are employed to select a single class of ions from memory crystal. The frequency of each pulse is swept over 5 MHz over 1 ms and these four pulses are repeated for 100 times. The second step is the so-called spin polarization process, where chirped pulses with center frequencies of $f_{15}$ and $f_{35}$ are applied to initialize these ions to the state $\left|\pm 5/2\right\rangle_g$. Finally, an absorption structure on the $\left|\pm 1/2\right\rangle_g$ state can be prepared by the backpump light with a center frequency of $f_{53}$. To achieve high efficiency for the control pulses, here we limit the bandwidth of the back pump light to 700 kHz to prepare an isolated absorption peak inside a transparent spectral range on the $\left|\pm 1/2\right\rangle_g$ state. Meanwhile, a chirped pulse at $f_{35}$ is used to empty the $\left|\pm 3/2\right\rangle_g$ state to be ready for spin-wave storage. In the preparation process, we also create a transparency window of ~1 MHz in the filter crystal, to transmit the signal at $f_{15}$. The absorption depth of the memory crystal after spectral preparation is $d = 0.6$. The detailed structures are presented in Supplementary Fig. 1a in the Supplementary Information.

**Noiseless photon echo**. The pulse sequence for NLPE is presented in Fig. 1b. The signal pulse at $f_{15}$ incidents at the time $t_0$. In the following, we denote $\pi$ pulse at the frequency of $f_{ij}$ as $\pi_{ij}$. The first $\pi_{35}$ pulse, applied at the time $t_1$, converts the optical excitation $\left|\pm 1/2\right\rangle_g \leftrightarrow \left|\pm 5/2\right\rangle_e$ into the spin coherence of $\left|\pm 1/2\right\rangle_g \leftrightarrow \left|\pm 3/2\right\rangle_g$. The first $\pi_{13}$ pulse, applied at time $t_2$, converts the spin coherence of $\left|\pm 1/2\right\rangle_g \leftrightarrow \left|\pm 3/2\right\rangle_g$ into the optical coherence of $\left|\pm 3/2\right\rangle_e \leftrightarrow \left|\pm 3/2\right\rangle_g$. The standard four-level photon echo[20] at $t_2 + t_1 - t_0$ is silenced due to a mismatch of wavevectors caused by the non-collinear configuration of the signal and control beams[18] (details in Supplementary Note 2). We note that, for implementations in the MW regime, this noisy echo can be silenced by dynamically detuning the cavity resonance[23,24]. The silenced echo can be recalled in a similar fashion to that in ROSE protocol[18]. At the time of $t_3$ and $t_4$, the second $\pi_{13}$ and the second $\pi_{35}$ are applied to double rephase the atomic ensemble and readout the echo from a non-inverted medium. In Supplementary Note 2, we provide a detailed model for the analysis of the NLPE protocol based on a complete quantum treatment on photon–atom interactions. The spatial phase-matching condition is

$$\mathbf{k}_{echo} = \mathbf{k}_0 - \mathbf{k}_1 - \mathbf{k}_2 + \mathbf{k}_3 + \mathbf{k}_4, \qquad (1)$$

where $\mathbf{k}_{echo}$ is the wavevector of the echo, and $\mathbf{k}_i$ represents the wavevector of each input pulse at time $t_i$, respectively. Since all the $\pi$ pulses have the same direction in the experiment, thus $\mathbf{k}_{echo} = \mathbf{k}_0$, resulting in an echo emission in the same direction as that of the input. The final echo emits at the time $t_5 = t_4 + t_3 - t_2 - t_1 + t_0$ and its frequency is $f_{15}$ which is the same as the input signal, in contrast to that in four-level photon echo.

Schematic of the experimental setup is shown in Fig. 2a. The laser is a frequency-doubled semiconductor laser at 580 nm, which is locked to an ultra-stable cavity to achieve a linewidth

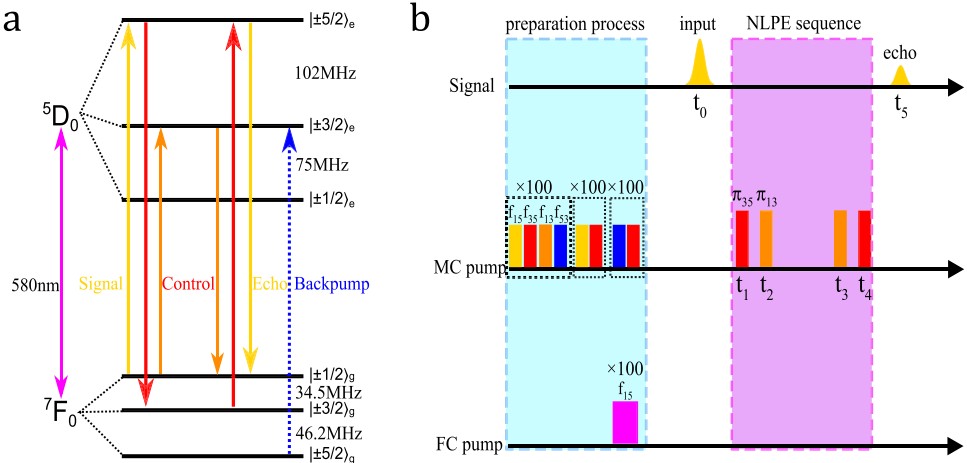

**Fig. 1 Energy level diagram and experimental sequence. a** Energy-level structure of $^{151}Eu^{3+}$ in a $Y_2SiO_5$ crystal at zero magnetic fields. The laser is resonant with $^5D_0$ and $^7F_0$ electronic states at 580 nm. Signal pulse (yellow) is resonant with $\left|\pm 1/2\right\rangle_g \leftrightarrow \left|\pm 5/2\right\rangle_e$, and control pulses are resonant with $\left|\pm 3/2\right\rangle_g \leftrightarrow \left|\pm 5/2\right\rangle_e$ (red) and $\left|\pm 1/2\right\rangle_g \leftrightarrow \left|\pm 3/2\right\rangle_e$ (orange), respectively. The back pump light (blue) which is resonant with $\left|\pm 5/2\right\rangle_g \leftrightarrow \left|\pm 3/2\right\rangle_e$ is employed in the preparation step for isolating a single class of ions in the inhomogeneously broadened ensemble. **b** Experimental pulse sequence. Including the signal sequence and the pump sequence for the memory crystal (MC) and filter pump sequence for filter crystal (FC). We use $f_{ij}$ to denote the center frequency of light, which is resonant with the atomic transition. The preparation process includes three steps: class cleaning, spin polarization, and back pump (see the text for details). After the preparation process, the signal pulse incidents at $t_0$, and control pulses $\pi_{35}, \pi_{13}, \pi_{13}, \pi_{35}$ incident at $t_1, t_2, t_3$, and $t_4$, respectively. Finally, the NLPE emits at $t_5 = t_4 + t_3 - t_2 - t_1 + t_0$.

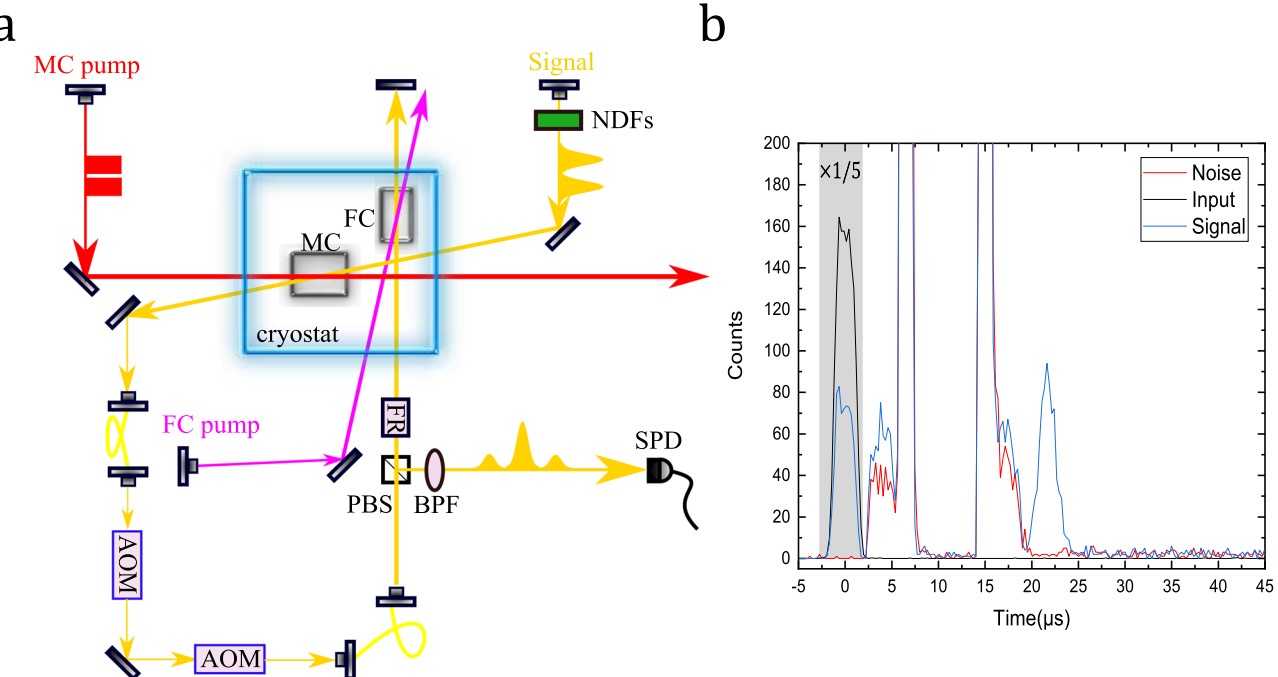

**Fig. 2 Experimental setup and photon counting histogram. a** Schematic of the experimental setup. The signal beam and the pump beam for the memory crystal (MC) counter propagate with each other and overlap on the MC with a small angle. The signal beam is attenuated to the single-photon level by neutral density filters (NFDs). The readout signal is collected by a single-mode fiber and passes through two acoustic-optic modulators (AOM) to shift the optical frequency by 400 MHz. The signal is spectrally filtered by the filter crystal (FC) and two 1-nm band-pass filters (BPF). The FC is doubled passed with the help of a polarization beam splitter (PBS) and a Faraday rotator (FR) and the signal is finally detected by a single photon detector (SPD). **b** Photon counting histogram for storage of weak coherent pulses with an average photon number of 1.17 ± 0.11 photons per pulse. The blue and red lines correspond to measurements with input and without input, respectively. The input (black line) is also included for reference. The input and signal data inside the shadowed area is minified by five times. The bin size is 262 ns, and the experiment is repeated for 50,000 trials.

below 1 kHz. The laser is split into three beams, the input mode, the pump mode for memory crystal, and the pump mode for the filter crystal. The non-collinear configuration of the signal beam and the counter-propagating control beam suppress noise in the spatial domain. Due to the different concentrations of $Eu^{3+}$ ions

in the memory crystal and the filter crystal, there is a center frequency difference of ~1 GHz for the optical absorption. To enhance the effective absorption of the filter crystal, two acoustic-optic modulators (AOM) are employed to shift the signal frequency by 400 MHz before entering the filter crystal.

## a

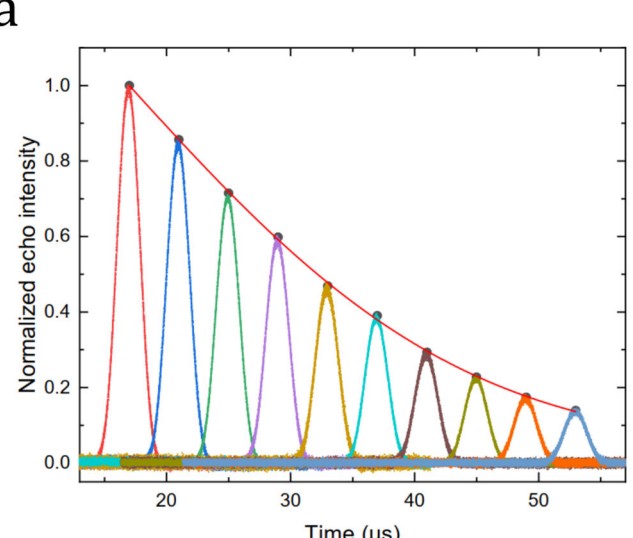

## b

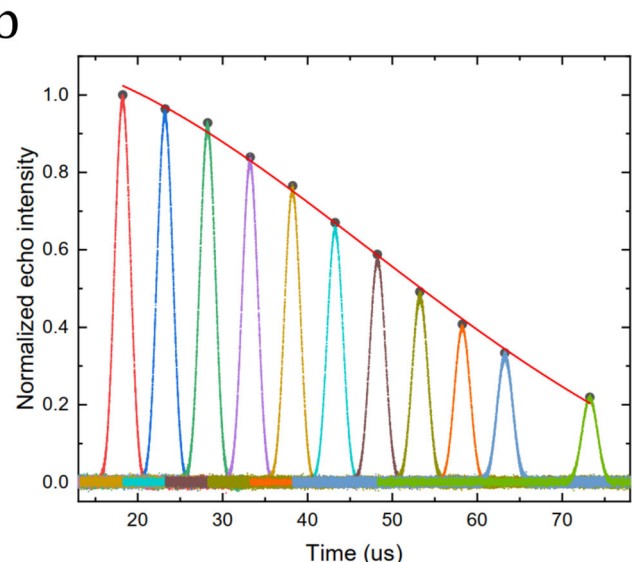

**Fig. 3 Efficiency decay in the NLPE memory.** Each data point represents the area of the readout echo over a 4-μs window. The echo traces are included for reference. **a** The decay of the echo intensity for varying delay $\tau_3 = 2(t_3 - t_2)$. **b** The decay of the echo intensity for varying delay $\tau_2 = t_4 - t_1$. The red lines are Gaussian fits based on Eq. (2).

Additionally, the filter crystal is double-passed to give an effective absorption depth of ~6.6.

According to the model presented in Supplementary Note 2, the total storage efficiency is

$$\eta_{\mathrm{NLPE}} = d^2 e^{-d} \cdot (\eta_{\mathrm{control}})^4 \cdot$$
$$e^{-\frac{\Gamma_{13}^2(t_4 - t_1)^2}{2\ln 2/\pi^2}} \cdot e^{-\frac{\Gamma_{\bar{3}\bar{5}}^2(t_3 - t_2)^2}{2\ln 2/\pi^2} - 2\gamma(t_3 - t_2)}, \tag{2}$$

where $\eta_{\mathrm{control}}$ is the average transfer efficiency of a single control pulse, $\Gamma_{13}$ and $\Gamma_{\bar{3}\bar{5}}$ are the inhomogeneous broadening of the spin transitions $\left|\pm 1/2\right\rangle_g \leftrightarrow \left|\pm 3/2\right\rangle_g$ and $\left|\pm 3/2\right\rangle_e \leftrightarrow \left|\pm 5/2\right\rangle_e$, respectively. $\gamma$ is the effective optical decoherence rates. The first item $d^2 e^{-d}$ defines the efficiency of a forward-retrieval echo, which is the standard form for all echo-based protocols[18,20,25]. The second item takes into account the efficiencies of four control pulses. The third and fourth items describe the dephasing during spin-wave storage and decoherence during optical storage.

The decay rates are different during spin-wave storage ($t_1 < t < t_2$ and $t_3 < t < t_4$) and optical storage ($t_2 < t < t_3$). We experimentally measure the efficiency decay using classical light as input. The decay curves of the echo intensity are measured with the delay times $\tau_3 = 2(t_3 - t_2)$ and $\tau_2 = t_4 - t_1$, as shown in Fig. 3a and b, respectively. The spin dephasing for $\tau_2 = t_4 - t_1$ can be well fitted by a Gaussian distribution with the parameter $\Gamma_{13} = 5.6 \pm 0.4$ kHz. This value is close to that estimated by spin-wave AFC storage (8 kHz)[26]. For $\tau_3 = 2(t_3 - t_2)$, the decay curve can be well fitted using the Gaussian function with the fitting parameter $\Gamma_{\bar{3}\bar{5}} = 18.6 \pm 2.5$ kHz with an estimated $\gamma$ of 12 kHz. The spin linewidth $\Gamma_{\bar{3}\bar{5}}$ is independently measured to be $20.2 \pm 0.5$ kHz by Raman–heterodyne detected nuclear quadrupole resonance. The large $\gamma$ of 12 kHz is presumably affected by the instantaneous spectral diffusion[18] due to the excitation of a large fraction of atoms to the excited state. Under the current experimental conditions, the theoretical efficiency is $\eta_{\mathrm{theo}} = 12.9\%$ assuming perfect control pulses. The experimentally measured efficiency is $\eta_{\mathrm{exp}} = 10.0\%$ with a storage time of 21.7 μs and the deduced efficiency of $\pi$ pulses is $\eta_{\mathrm{control}} = 93.8\%$. This storage efficiency is obtained using a sample with a weak absorption ($d = 0.6$). Higher efficiency can be obtained with a sample with large absorption and unit efficiency can obtain through special phase-matching configuration[18] or an impedance-matched cavity[27].

Now we implement the NLPE memory with single-photon-level inputs. The input signal is weak coherent light which is a truncated Gaussian pulse with a full-width at half-maximum of 2.62 μs, and the center of the pulse is $t_0 = 0$ μs. $\pi_{35}$ with a pulse length of 3.75 μs incidents after the input and the center of the pulse is $t_1 = 4.1$ μs. $\pi_{13}$ with a pulse length of 1.5 μs incidents at $t_2 = 6.6$ μs. Then we wait for 7 μs to separate the fourth $\pi$ pulse and the echo, the second $\pi_{13}$ pulse and the second $\pi_{24}$ pulse incident at $t_3 = 15.0$ μs and $t_4 = 17.4$ μs, respectively. All $\pi$ pulses are complex hyperbolic secant pulses to achieve a high robustness[18] and the parameters have been optimized according to the echo efficiency. The echo emits at $t_5 = t_4 + t_3 - t_2 - t_1 + t_0 = 21.7$ μs. The photon-counting histogram for storage of weak coherent pulses with an average photon number of $1.17 \pm 0.11$ photons per pulse is shown in Fig. 2b. The blue and red lines correspond to measurements with input and without input, and the input (black line) is included for reference. The storage efficiency of the NLPE echo is $10.0 \pm 0.4\%$. If we limit the readout signal in a window of 1.57 μs width then the efficiency is $6.4 \pm 0.2\%$ and the achieved SNR is $42.5 \pm 7.5$, as defined by $\mathrm{SNR} = \frac{S}{N}$, where $N$ is the noise counts without input and $S$ is the counts with input excluding noise counts.

**Analysis of residual noise in NLPE.** The noise during the light–matter interaction (such as the quantum memory considered here) can be categorized into two parts: coherent noise and incoherent spontaneous emission noise. The coherent noise (such as free induction decay) has the same mode as that of the strong rephasing pulse so it can be completely filtered out in principle[20]. On the contrary, the spontaneous emission noise is thought to be impossible to separate from the signal[7,18,19,23–25]. In all previous PE protocols, the echo is generated from an excited state which is the same one that is connected to the populated ground state with the rephasing $\pi$ pulses. Therefore, any remaining population in the excited state will produce indistinguishable spontaneous emission noise into the echo mode. As a result, the PE storage of light field is limited to ~14 photons at least[19]. In the NLPE protocol, the control pulses ($\pi_{13}$) which are applied on the populated ground state $\left|\pm 1/2\right\rangle_g$ will bring the population to the excited state $\left|\pm 3/2\right\rangle_e$ which is different from

the excited state ($|\pm 5/2\rangle_e$) that generates the echo. The spontaneous emission noise in the NLPE has a different frequency to the echo mode, which can be easily removed by a frequency filter such as the filter crystal employed in the current work. This observation can be confirmed by the data presented in Fig. 2b, where the noise after the first pair of $\pi$ pulses (where the medium is completely excited) is close to that after two pairs of $\pi$ pulses (where the medium is restored to the ground state). Imperfections of the control pulses will not introduce indistinguishable spontaneous emission noise to the final echo and this is the essential advantage compared to all previous PE protocols. We conclude that the spontaneous emission noise cannot be removed in a two-level system[7] or a three-level system[28], but it can be completely suppressed in a four-level system.

Although NLPE is free from any noise in principle, there is some residual noise in the actual implementation of NLPE memory as shown in Fig. 2b. The strongest spontaneous emission noise originated from the excited state $|\pm 3/2\rangle_e$ is approximately $e^d - 1$ after the first pair of $\pi$ pulses since the population is completely excited to $|\pm 3/2\rangle_e$ state[29]. After the filter crystal, the remaining noise is estimated by $(e^d - 1) * e^{-d_{FC}} = 1.1 * 10^{-3}$. According to the data presented in Fig. 2b, the measured noise after the first pair of $\pi$ pulses is $\sim 9 \times 10^{-4}$ photons per pulse, which is close to the expected spontaneous emission noise solely from the excited state $|\pm 3/2\rangle_e$. Therefore, we expect that the coherent noise is negligible in the current experiment. The spontaneous emission noise from the $|\pm 3/2\rangle_e$ should be strongly suppressed after the second pair of $\pi$ pulses since the population is brought back to the ground state.

Other spontaneous emission noise can be caused by the residual population in the excited state $|\pm 5/2\rangle_e$. Two processes can contribute to this noise. The first one is caused by the residual population of state $|\pm 3/2\rangle_g$ after spectral initialization. In this case, the spontaneous emission noise after the first pair of control pulses (where the residual ions initially at $|\pm 3/2\rangle_g$ are almost completely excited) should be much stronger than that after the second pair of control pulses (where the population is almost completely in the ground state). Since this kind of noise contributes to the noise after the first pair of control pulses, it should be negligible according to the analysis presented above.

The second one is caused by the spontaneous decay from the populated excited state $|\pm 3/2\rangle_e$ to the ground state $|\pm 3/2\rangle_g$ between the second and third control pulse. This process only brings the noise to the time window after the second pair of $\pi$ pulses because the final $\pi_{35}$ pulse can lift these ions to $|\pm 5/2\rangle_e$ and produce indistinguishable spontaneous emission noise. According to the data presented in Fig. 2b, the noise after the second pair of $\pi$ pulses is $\sim 1.5 \times 10^{-3}$ photons per pulse. In principle, such noise can be reduced to an acceptable level, using countermeasures such as employing a four-level system with appropriate selection rules to forbid the decay path of $|\pm 3/2\rangle_e$ to $|\pm 3/2\rangle_g$, and reducing the excited state storage time to a small value as compared to the excited state lifetime to avoid too much decay during the storage in the excited state. In practice, the short excited-state storage time would put a limit on the temporal multimode capacity of the NLPE protocol.

**Fidelity of qubit storage**. To further characterize its compatibility in quantum information storage, here we assess the memory performance by the fidelity of qubit storage. We employ the time-bin encoded qubit since it is particularly robust for long-distance transmission[30]. We use $|e\rangle$ and $|l\rangle$ to denote the early

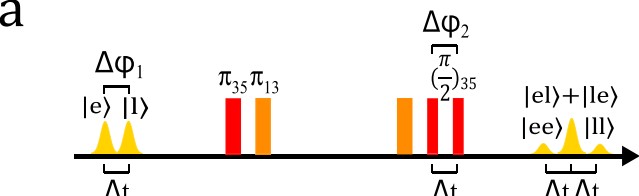

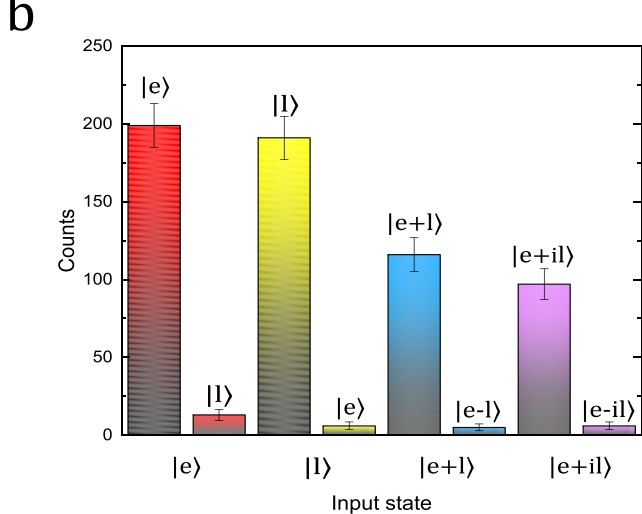

**Fig. 4 Time-bin qubits fidelity measurement. a** The pulse sequence for the preparation and analysis of time-bin qubits. The time difference between $|e\rangle$ and $|l\rangle$ is $\Delta t$, and the relative phase is controlled as $\Delta\varphi_1$. After three $\pi$ pulses that are the same as that required in standard NLPE, two $(\frac{\pi}{2})_{35}$ are employed to read out each input twice. The time difference between two $(\frac{\pi}{2})_{35}$ is also $\Delta t$, and the relative phase is controlled as $\Delta\varphi_2$. **b** Storage performances for four input qubit states. The average photon number per qubit is 2.29 photons. For input states $|e\rangle$ and $|l\rangle$, the output is projected on $|e\rangle$ and $|l\rangle$. For input superposition states ($|e\rangle + |l\rangle$ and $|e\rangle + i|l\rangle$), the output is projected superposition states, which generate the maximal interference and minimal interference results in the middle output bin. Photon-counting histograms can be found in Supplementary Fig. 2 and the experiments are repeated for 20,000 trials. The error bars represent one standard deviation of photon counts assuming Poissonian photon statistics.

time bin and the late time bin, respectively. $|e\rangle$ and $|l\rangle$ are separated by a delay of $\Delta t$ and the relative phase is controlled to be $\Delta\varphi_1$. The memory performance is assessed by four different input states $|e\rangle$, $|l\rangle$, $|e\rangle + |l\rangle$, and $|e\rangle + i|l\rangle$, and the average photon number ($\mu$) is $2.29 \pm 0.06$ per input qubit. The fidelity of the input state $|e\rangle$ is defined by $F_e = \frac{S+N}{S+2N}$. Here $N$ is the noise counts in the late time bin and $S$ is the signal counts excluding noise counts in the early time bin. The fidelity $F_l$ for $|l\rangle$ is defined in a similar fashion. Here the spacing between the two $\pi_{13}$ pulses is set as 13 $\mu$s to separate the final $\pi$ pulse and the first echo.

For states $|e\rangle + |l\rangle$ and $|e\rangle + i|l\rangle$, one will require an unbalanced Mach–Zehnder interferometer for measurements on the superposition states. Here we take advantage of the memory itself to serve as a temporal beam splitter for the unbalanced Mach–Zehnder interferometer. The scheme for the preparation and analysis of superposition time-bin qubits is presented in Fig. 4a. The last control pulse $\pi_{35}$ in the NLPE protocol is divided into two $\frac{\pi}{2}$ pulses at $f_{35}$ ($(\frac{\pi}{2})_{35}$) pulses with the pulse delay of $\Delta t$ which is the same as the delay of two input bins. In this way, each

| Table 1 Storage fidelity of time-bin qubits with $\mu = 2.29$. | | | | |
|---|---|---|---|---|
| | $|e\rangle$ | $|l\rangle$ | $|e\rangle + |l\rangle$ | $|e\rangle + i|l\rangle$ |
| Fidelity | 93.9 ± 1.6% | 97 ± 1.2% | 95.9 ± 1.8% | 94.2 ± 2.3% |

input can be read out for two times and the memory has three outputs: $|ee\rangle$, $|el\rangle + |le\rangle$ and $|ll\rangle$. An interference presents in the middle of the readout. For the input state $|e\rangle + |l\rangle$, we change the relative phase $(\Delta\varphi_2)$ of the two $(\frac{\pi}{2})_{35}$ pulses to find the maximal value $C_{\max}$ and minimum value $C_{\min}$ of the middle readout, and then calculate the visibility $V = \frac{C_{\max} - C_{\min}}{C_{\max} + C_{\min}}$. The storage fidelity for input states $|e\rangle + |l\rangle$ is $F_+ = \frac{V+1}{2}$. The storage fidelity $F_-$ for $|e\rangle + i|l\rangle$ is defined in a similar form. The measured results are presented in Fig. 4b and Table 1. Finally, the average fidelity is $F = \frac{1}{3}F_{el} + \frac{2}{3}F_{+-} = 95.2 \pm 1.8\%$ for $\mu = 2.29$, where $F_{el}$ is the mean value of $F_e$ and $F_l$, and $F_{+-}$ is the mean value of $F_+$ and $F_-$[30]. In Supplementary Note 1, we calculate the maximal fidelity that can be achieved using the classical measure-and-prepare strategy, taking into account the finite storage efficiency and the Poisson distribution of the photon source[30–32]. The measured fidelity is well above the classical limit of 88.0% at $\mu = 2.29$, unambiguously demonstrating the NLPE memory operates in the quantum domain.

## Discussion

The application demonstrated in this work is an NLPE optical quantum memory in a $Eu^{3+}$:$Y_2SiO_5$ crystal. As a PE optical memory, our results reduced the noise by 670 times as compared to that of the previous demonstration based on ROSE[19]. We have further implemented the ROSE protocol in our system, the measured noise inside the detection window is 0.046 photons per trial which is more than 30 times larger than the noise of NLPE with the same experimental configurations, directly indicating the advantages of NLPE in noise suppression.

To date, AFC is the only protocol that has enabled qubit storage using spin-wave excitation in rare-earth-ion-doped materials[13,30,31]. It is instructive to compare the performances of these two protocols. Due to the limited storage time in the excited state and the bandwidth limit caused by the instantaneous spectral diffusion[18], NLPE has a lower temporal multimode capacity as compared to that of AFC. However, without a cavity enhancement configuration, the sample absorption and the storage efficiency are significantly reduced due to the complex spectral tailoring in the AFC memory[13]. For the sample absorption considered here ($d = 0.6$), the optimal storage efficiency of AFC would be limited to 2.7% (details in Supplementary Note 1). NLPE solves this problem by insisting on the direct optical rephasing and the efficiency of NLPE obtained here is more than three times larger than that can be achieved with AFC. In practice, the spin-wave AFC would have a storage efficiency much lower than that of the optimal two-level AFC considered here. As a result, the achieved SNR of NLPE is enhanced by four times as compared to that obtained with spin-wave AFC using the same material ($Eu^{3+}$:$Y_2SiO_5$)[15]. The high-efficiency property of NLPE is crucial for applications in the zero-first-order-Zeeman magnetic field[21], where the sample absorption is severely limited[17], to achieve optical quantum storage over several hours. Its high fidelity, high efficiency, and potentially long lifetime, should enable the construction of quantum repeaters[5,6] and transportable quantum memories[17], for the ultimate goal of long-distance quantum communications.

Similar to the standard PE, no spectral tailoring is required in NLPE. Therefore, it is inherently suitable for implementations in various atomic and molecular systems with extended working frequencies, which may stimulate novel applications of the ultralow-noise PE across many disciplines.

## Data availability

The data that support the findings of this study are available from the corresponding author upon reasonable request.

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

## Acknowledgements
This work is supported by the National Key R&D Program of China (No. 2017YFA0304100), the National Natural Science Foundation of China (Nos. 11774331, 11774335, 11504362, 11821404, and 11654002), Anhui Initiative in Quantum Information Technologies (No. AHY020100), the Key Research Program of Frontier Sciences, CAS (No. QYZDY-SSW-SLH003), the Science Foundation of the CAS (No. ZDRW-XH-2019-1) and the Fundamental Research Funds for the Central Universities (Nos. WK2470000026, WK2470000029). Z.-Q.Z. acknowledges the support from the Youth Innovation Promotion Association CAS.

## Author contributions
Y.-Z.M., M.J., and D.-L.C. contributed equally. Z.-Q.Z. designed the NLPE protocol; Z.-Q.Z. and C.-F.L. conceived the experiment; Y.-Z.M. and M.J. performed the experiment and analyzed the data, D.-L.C. and Y.-Z.M. constructed the theoretical model; Y.-Z.M., D.-L.C. and Z.-Q.Z. wrote the manuscript; Z.-Q.Z., C.-F.L. and G.-C.G. supervised the project.

## Competing interests
The authors declare no competing interests.
