## [Peer Review File · Nature Communications]

Reviewers' Comments:

Reviewer #2:

Remarks to the Author:

I think that in the revised manuscript the authors have satisfactorily addressed most of the major concerns raised by me and the other referees. The manuscript is significantly improved by clarifying the inherent advantages and disadvantages of the proposed protocol. The direct performance comparison with respect to the AFC and ROSE approaches is quite useful for readers.

On the other hand, still I disagree with the proposed title of the manuscript. As the authors stated in the revised paper, using this protocol it is possible to reach an "acceptable level" of photonic noise under certain conditions. In particular, the noise is strongly coupled to the lifetime of the relevant optical transition, which is one of the most fundamental parameters, governing the atomic dynamics. Also, in general, there is a trade off between the noise level and multimode storage capacity.

Considering these factors plus lack of the experimental evidence for ultra-low noise levels, I feel that the title of "noiseless photon echo" is too broad and too strong for the current proposal and demonstration. This title also gives a wrong impression of that all other photon-echo memory techniques (like AFC) are noisy.

In this sense, this study achieves the suppression of noise for a certain class of photon echoes, relying on optical rephasing. For this reason, I would suggest a title like "suppression of noise in optically rephased photon-echoes" or "Elimination of noise for optically rephased photon-echo quantum memories"

Other than this point, I recommend this manuscript for publication in Nature Communications.

In the following we provide a point-to-point response to the reviewer's comments. All the original comments are in black font and our reply is in blue font.

Reviewer #2 (Remarks to the Author):

I think that in the revised manuscript the authors have satisfactorily addressed most of the major concerns raised by me and the other referees. The manuscript is significantly improved by clarifying the inherent advantages and disadvantages of the proposed protocol. The direct performance comparison with respect to the AFC and ROSE approaches is quite useful for readers.

We thank the reviewer for these positive comments.

On the other hand, still I disagree with the proposed title of the manuscript. As the authors stated in the revised paper, using this protocol it is possible to reach an "acceptable level" of photonic noise under certain conditions. In particular, the noise is strongly coupled to the lifetime of the relevant optical transition, which is one of the most fundamental parameters, governing the atomic dynamics. Also, in general, there is a trade off between the noise level and multimode storage capacity. Considering these factors plus lack of the experimental evidence for ultra-low noise levels, I feel that the title of "noiseless photon echo" is too broad and too strong for the current proposal and demonstration. This title also gives a wrong impression of that all other photon-echo memory techniques (like AFC) are noisy.

AFC is a noiseless memory protocol and the achieved noise level is similar as that we obtained here with NLPE. To be rigorous, AFC is an echo-based protocol but it is not a "photon echo". By definition, photon echo (spin echo) refers to the refocusing process which is induced by an optical rephrasing pulse (https://en.wikipedia.org/wiki/Spin_echo). As a result, our original title implies that all other photon echoes are noisy but it doesn't imply that other echo-based protocols (like AFC) are noisy.

In this sense, this study achieves the suppression of noise for a certain class of photon echoes, relying on optical rephasing. For this reason, I would suggest a title like "suppression of noise in optically rephased photon-echoes" or "Elimination of noise for optically rephased photon-echo quantum memories"

Nonetheless, to avoid any possible confusions, we follow the suggestion of the reviewer and modify the title to "A noiseless photon echo with optical rephrasing".

Other than this point, I recommend this manuscript for publication in Nature Communications.

Thank you for the recommendation.